# A Characterization and Functional Analysis of Peroxisome Proliferator-Activated Receptor Gamma Splicing Variants in the Buffalo Mammary Gland

**DOI:** 10.3390/genes15060779

**Published:** 2024-06-13

**Authors:** Shuwan Wang, Honghe Ren, Chaobin Qin, Jie Su, Xinhui Song, Ruijia Li, Kuiqing Cui, Yang Liu, Deshun Shi, Qingyou Liu, Zhipeng Li

**Affiliations:** 1Guangxi Key Laboratory of Animal Reproduction, Breeding and Disease Control, College of Animal Science and Technology, Guangxi University, Nanning 530004, China; wswan@st.gxu.edu.cn (S.W.); 2018301027@st.gxu.edu.cn (H.R.); 2018401005@st.gxu.edu.cn (C.Q.); 18438591697@163.com (J.S.); songxinhui@st.gxu.edu.cn (X.S.); 2018302017@st.gxu.edu.cn (R.L.); ardsshi@gxu.edu.cn (D.S.); 2Guangdong Provincial Key Laboratory of Animal Molecular Design and Precise Breeding, School of Life Science and Engineering, Foshan University, Foshan 528225, China; kqcui@fosu.edu.cn (K.C.); qyliu-gene@fosu.edu.cn (Q.L.); 3Guangxi Zhuang Autonomous Region Center for Analysis and Test Research, Nanning 530022, China

**Keywords:** PPARG, buffalo mammary epithelial cells, fatty acid synthesis, gene regulation

## Abstract

Peroxisome proliferator-activated receptor γ (PPARG) has various splicing variants and plays essential roles in the regulation of adipocyte differentiation and lipogenesis. However, little is known about the expression pattern and effect of the PPARG on milk fat synthesis in the buffalo mammary gland. In this study, we found that only *PPARG-X17* and *PPARG-X21* of the splicing variant were expressed in the buffalo mammary gland. Amino acid sequence characterization showed that the proteins encoded by *PPARG-X17* and *PPARG-X21* are endonuclear non-secreted hydrophilic proteins. Protein domain prediction found that only the *PPARG-X21*-encoded protein had PPAR ligand-binding domains (NR_LBD_PPAR), which may lead to functional differences between the two splices. RNA interference (RNAi) and the overexpression of *PPARG-X17* and *PPARG-X21* in buffalo mammary epithelial cells (BMECs) were performed. Results showed that the expression of fatty acid synthesis-related genes (*ACACA*, *CD36*, *ACSL1*, *GPAT*, *AGPAT6*, *DGAT1*) was significantly modified (*p* < 0.05) by the RNAi and overexpression of *PPARG-X17* and *PPARG-X21*. All kinds of FAs detected in this study were significantly decreased (*p* < 0.05) after RNAi of *PPARG-X17* or *PPARG-X21*. Overexpression of *PPARG-X17* or *PPARG-X21* significantly decreased (*p* < 0.05) the SFA content, while significantly increased (*p* < 0.05) the UFA, especially the MUFA in the BMECs. In conclusion, there are two *PPARG* splicing variants expressed in the BMECs that can regulate FA synthesis by altering the expression of diverse fatty acid synthesis-related genes. This study revealed the expression characteristics and functions of the *PPARG* gene in buffalo mammary glands and provided a reference for further understanding of fat synthesis in buffalo milk.

## 1. Introduction

Fatty acids (FAs) are important nutrients and growth factors [1], contributing to immune system health and participating in metabolism, serving as a source of cellular energy [2]. There are approximately 400 different types of FAs in cow milk, making it one of the most complex sources of natural fats [3]. The content and composition of FAs are important indicators of milk flavor, nutritional value, and economic value. Buffalo milk contains more FAs than the milk from cows, goats, and sheep [4], especially unsaturated fatty acids (UFA), such as linoleic acid, linolenic acid, and arachidonic acid, which are beneficial to human health [5,6]. Revealing the molecular mechanism that regulates FA synthesis and metabolism in buffalo mammary epithelial cells (BMECs) can provide an essential reference for the development and application of buffalo’s excellent lactation traits and breeding of cows.

There are two main sources of FAs in milk, one is directly derived from blood, and the other is de novo synthesis in breast cells [7]. Studies have found that in early lactation, the FAs in milk are mainly taken up by the blood [7,8]. The synthesis of FAs from acetic acid and butyric acid by mammary epithelial cells begins in the second week of lactation and peaks on day 30 [7,9]. Therefore, 57.8% of FAs in milk are derived from de novo synthesis by mammary epithelial cells from the second month of lactation [7]. Gene expression regulation is one of the most important factors affecting FA synthesis and metabolism, and Peroxisome Proliferator-Activated Receptor γ (PPARG) is a key candidate gene in regulating lipid storage and metabolism in cells [10]. It is a nuclear receptor and contains three subtypes: PPARG-α, PPARG-β, and PPARG-γ [11]. The target genes of *PPARG* include many genes related to FA transport, such as Lipoprotein Lipase (*LPL*), Cluster of Differentiation 36 (*CD36*), and Acyl-CoA Synthetase Long-Chain Family Member 1 (*ACSL1*) [12]. PPARG can also regulate the metabolism of FAs and lipids by regulating the expression of genes related to the de novo synthesis of FAs, such as fatty acid synthase (*FASN*) and acetyl-CoA carboxylase α (*ACACA*). Studies have found that all three subtypes of PPARGs can bind to FAs and have a general preference for long-chain polyunsaturated FAs (PUFAs) (<10 μM to PPAR-α, 10–100 μM to PPAR-β and PPAR-γ) [13]. Buffalo mammary glands have been found expressing two protein isomers and eight splices of *PPARG*, and these variable splicing variants may be related to the post-transcriptional regulation of lactation [14]. The expression of *PPARGC1 A* increased 11-fold after 120 days of lactation, but the expression of *PPARGC1 B* continued to decline throughout lactation, indicating that *PPARGC1 A* plays an important role in cow milk fat synthesis [7]. However, these studies only focused on the expression pattern of *PPARG* in mammary glands. Little is known about the effects of *PPARG* on FA synthesis and regulatory functions in buffalo mammary cells.

This study aims to systematically analyze and identify the expression pattern and molecular characteristics of *PPARG* in BMECs, reveal its influence on the content and composition of fatty acid synthesis in BMECs using RNA interference and gene overexpression, and preliminarily analyze its molecular regulatory mechanism. This study will provide a further understanding of the fatty acid synthesis mechanism of BMECs and provide references for the development and application of buffalo lactation-related genetic resources.

## 2. Materials and Methods

### 2.1. Experimental Animals and Sampling

Murrah buffalo mammary gland tissues used in this study were collected at the Buffalo Breeding Farm of the Buffalo Research Institute, Chinese Academy of Agricultural Sciences, Nanning, Guangxi, China. The buffalo mammary gland tissues were collected 2 times on the farm during the mid-lactation period. Buffalo milk was also collected during early (30–100 days), mid (100–200 days), and late (>200 days) lactation manually by the farmers. All of the selected buffaloes were in the second or third parity, with ages between 5 and 8 years. The milk samples were collected in summer (June–July), between 5:00 and 6:00 a.m., on the same day for each lactation stage. The breasts were cleaned with warm water, and the region around the teat was sterilized with 75% ethanol. A few streams of foremilk were discarded, and 100 mL of milk from each buffalo was collected in sterile vials. The samples were frozen immediately in liquid nitrogen to transport to the laboratory and were stored at −80 °C until further use.

### 2.2. Milk Routine Component Analysis

We collected 30 samples from each lactation period, and each milk sample was collected in triplicate. The composition (lactose, fat, protein, total solid, and non-fat solid) was analyzed by a multifunction analyzer for dairy products (MilkoScan FT-120, FOSS Electric A/S, Hillerod, Denmark). 

### 2.3. Extraction and Component Analysis of Fatty Acids (FA in Milk and BMECs)

Milk fat extraction and gas chromatography analysis were performed following the method reported before [5]. In brief, 2 grams of milk sample was mixed with 0.4 mL ammonia 25%, 1 mL ethyl alcohol (95%), and 5 mL hexane, vortexed and centrifuged at 1600× *g* at 4 °C. The upper layer was collected, and a second extraction with 1 mL ethyl alcohol (95%) and 5 mL hexane was performed. A third extraction was made by using 5 mL of hexane. The extracted fat was dried, weighed, and finally dissolved in hexane. FA composition was determined by gas chromatography using a Shimadzu GC 2014 C (Kyoto, Japan) gas chromatograph equipped with an FID and a capillary column (Agilent DB23, Santa Clara, CA, USA; 30 m × 0.32 mm i.d.; film thickness 0.25 μm). The carrier gas was kept in high-purity nitrogen, and the injection volume was 1 μL. Individual FA methyl esters were identified by comparing them to a standard mixture of 37 Component FAME Mix (Supelco, Bellefonte, PA, USA). A nonadecanoic acid was used as an internal standard to increase the veracity of the peak normalization. 

To detect the FA component in the BMECs, the BMECs transfected with siRNAs or overexpression vectors were collected in a 100 mL colorimetric tube. A solution containing 2 mL of 95% ethanol, 4 mL of water, and 10 mL of 8.3 mol/L hydrochloric acid was added to the tube. The sample was extracted 3 times using a mixture of petroleum ether and ether. The combined extract was transferred into a new flask. The subsequent processes are the same as the method described above. For all studied FAs, the coefficient of variation [(SD/mean) × 100] was <3.5%, suggesting good repeatability of GC data. All milk FA compositions were expressed as mg per 100 g of fat. 

### 2.4. Extraction of Total RNA and DNA from Buffalo Milk

Freshly filtered buffalo milk was centrifuged at 4 °C and 1500 rpm for 10 min to collect the cell pellet. Total RNA was extracted using the TRIzol method. In brief, the cell pellet was harvested and mixed thoroughly with 1 mL of TRIzol™ LS solution (Invitrogen Life Technologies Inc., Carlsbad, CA, USA) using a vortex (Thermo Fisher Scientific, Waltham, MA, USA). The lipids were removed from the top layer after the centrifugation (Thermo Fisher Scientific, Waltham, MA, USA). Chloroform (Sigma-Aldrich, St. Louis, MO, USA) was used to separate the aqueous phase, and isopropanol (Sigma-Aldrich, MO, USA) was used to precipitate the RNA. The RNA was washed 2 times with 75% ethanol and dissolved with DEPC water (Beyotime, Shanghai, China). RNA concentration and integrity were detected using the NanoDrop 2000 (Thermo Fisher Scientific, Waltham, MA, USA) and agarose gel electrophoresis. To synthesize the cDNA, the genomic DNA was removed using DNase treatment. First-strand cDNA was then synthesized using the RevertAid First Strand cDNA Synthesis Kit (K1622, Thermo Fisher Scientific, USA). The single-strand cDNA obtained was stored at −20 °C. 

For DNA extraction, the cell pellet was mixed with a 200 μL digestion solution, containing 10 mM Tris-Cl (pH 8.0), 15 mM NaCl, 10 mM EDTA (pH 8.0), 0.4% SDS, and 200 μg/mL proteinase K (Solabio, Beijing, China), in distilled water. The samples were then incubated in the water at 55 °C for 1 h. Then, the samples were mixed with a DNA extraction solution, containing 0.5 volume Phenol reagent for DNA extraction (Solabio, Beijing, China) and 0.5 volume chloroform/isoamyl alcohol (24:1). Mix thoroughly for 10 s and leave on ice for 15 min. Centrifuge the samples at 4 °C and 1500 rpm for 10 min, collect the water phase (upper layer), and add an equal volume of isopropyl alcohol to precipitate DNA; wash with 75% ethanol twice, slightly dry, and dissolve in the appropriate amount of TE buffer for further use.

### 2.5. Identification of PPARG Splices from Buffalo Somatic Cells in Milk 

There are 21 *PPARG* transcript variants with 4 non-frameshifting indels in the buffalo genomes based on the NCBI database. Primers targeted to the specific region of each *PPARG* variant for PCR were designed to identify the *PPARG* splices involved in the buffalo mammary gland. Primers targeted to the whole CDS sequences for RT-PCR were designed to colon the *PPARG* from the cDNA of buffalo somatic cells in milk. All of the primers used in this study were designed using Oligo 7 software (Appendix A) and synthesized by Beijing Qingke Biotechnology Co., Ltd. PCR was performed using Q5 2 × Master Mix (NEB ENGLAND BioLabs Inc., Ipswich, MA, USA). RT-PCR was performed using HiScript II One Step RT-PCR Kit (Dye Plus) (Vazyme Biotech Co., Ltd., Nanjing, China). 

### 2.6. Characteristic Analysis of the PPARG Isoforms

Characteristic analysis of the PPARG isoforms was performed referring to the previous reports [15,16]. In brief, a neighbor-joining phylogenetic tree was constructed in MEGA11 (v11.0.11). The common characteristics of the two *PPARG* isoforms were predicted using the ProtParam tool. Motif analysis was performed using the MEME Suite tool. The conserved domains of the *PPARG* isoforms were analyzed using the NCBI conserved domain database and CD-Search tool. TB tools (v1.098745) were used to integrate the results of the motif and the protein conserved domain. The amino acid sequences of *PPARG* isoforms were submitted to the online server SWISS-MODEL to obtain the tertiary structure of each protein, and the secondary structure feature fold recognition end homology modeling was identified. 

### 2.7. Isolation, Culture, and Purification of Buffalo Mammary Epithelial Cells

The buffalo mammary epithelial cells (BMECs) were cultured and purified as reported previously [17]. Briefly, fresh buffalo mammary gland tissue was obtained from the butchery and washed three times with normal saline (0.9% NaCl). The acinus portion was extracted from the mammary gland tissue, washed with normal saline (0.9% NaCl), and transferred into high-resistance PBS (containing 400 IU mL^−1^ each of penicillin and streptomycin) until it was brought back to the laboratory. Then, the tissue pieces were placed in culture dishes on a clean bench, cut into 1–2 mm pieces, tiled on the bottom of the culture dish, and cultured in an incubator (38.5 °C) for 4 h. The tissue pieces were then inverted and cultured with F12/DMEN (Gibco, Waltham, MA, USA), which contained 20% serum (Gibco, Waltham, MA, USA), in the upright position overnight. Once the epithelial cells started growing after approximately 8 days, they were isolated through trypsin digestion combined with a cell adherence speed method. The purification procedure was repeated three times, and BMECs at 3–4 generations in the subculture were used for the following studies. The cells were cultured with F12/DMEN (Gibco, Waltham, MA, USA), which contained 20% serum (Gibco, Waltham, MA, USA), with a cell incubator (Thermo Fisher Scientific, Waltham, MA, USA). 

### 2.8. siRNA Synthesis and Overexpression Vector Construction of Buffalo PPARG-X17 and PPARG-X21

siRNAs targeting *PPARG-X17* and *PPARG-X21* were designed and synthesized by Sangon Biotech (Shanghai, China) with a control sequence (Appendix A). *PPARG-X17* and *PPARG-X21* were inserted into the pcDNA3.1-eGFP vector to construct the respective overexpression vectors. 

### 2.9. Transfection of BMECs

Transfection of siRNAs and overexpression vectors was performed using Lipofectamine™ 2000 (Invitrogen, USA) according to the manufacturer’s instructions. The overexpression vectors or siRNAs were added to the transfection reagent at a 1:2 ratio after the cell density reached 80%. Fluorescence was detected to determine the transfection efficiency 24 h after the transfection. The cells were collected 48 h after the transfection, and qRT-PCR was performed to analyze the gene expression. Each transfection experiment was performed three times, and each sample was detected three times. Cells transfected with the pcDNA3.1-eGFP vector or random siRNA sequences were used as the negative control.

### 2.10. qRT-PCR Analysis

Primers for qRT-PCR were designed using Oligo 7.0 software (Appendix A) to determine the expression level of *PPARγ-X17* and *PPARγ-X21*. qRT-PCR was performed using the SYBR qPCR Master Mix (Vazyme Biotech Co., Ltd., Nanjing, China) following the manufacturer’s instructions. Fluorescence data were acquired using the fluorescence ratio PCR instrument (Roche, Shanghai, China). More than three biological and technical replicates were maintained. The relative gene expression was calculated using the 2^−ΔΔCT^ method, and *b-actin* served as the reference gene.

### 2.11. Statistical Analysis

Statistical analysis was performed using analysis of variance (ANOVA) with Duncan’s multiple range (DMR) test in SPSS version 23.0 software (IBM SPSS Statistics, New York, NY, USA) to analyze the differences in gene expression and fatty acid content. Data were expressed as mean ± SEM, and *p* < 0.05 was considered statistically significant.

## 3. Results

### 3.1. Composition Analysis of Milk and FAs from Different Lactation Periods

Table 1 shows that the fat in early lactation is significantly lower than that in middle and late lactation (*p* < 0.05). The amounts of protein and total solids (TSs) in early lactation are significantly lower than those in late lactation (*p* < 0.05). No significant difference was found in the content of lactose and non-milk solids between different lactation periods. The composition of the FAs in the milk samples from different lactation periods was further analyzed (Table 2). Results showed that the total FA (TFA) content in middle and late lactation was significantly higher than that in early lactation (*p* < 0.05), which is consistent with the routine analysis shown above. Among the FAs, C16:0 was the most abundant, accounting for 30.2%, 27.6%, and 29.7% of the total FAs in the three periods, respectively. The content of unsaturated fatty acid (UFA), especially monounsaturated fatty acid (MUFA), is significantly increased (*p* < 0.05) as lactation progresses. UFAs account for 33.9%, 34.7%, and 39.5% of the total FAs, and MUFAs account for 23.9%, 24.4% and 27.9% of the total FAs in the three periods, respectively. Docosahexaenoic acid (DHA) content is highest in the milk from early lactation (2.1 ± 0.5) and then gradually decreases in mid-lactation (1.7 ± 0.4), and it is not in milk from late lactation. Long-chain fatty acids (LCFAs) seem to be the main increased FA (*p* < 0.05) as lactation progresses (Table 1). 

### 3.2. Identification of PPARG Splices from Buffalo Somatic Cells in Milk

Primers targeted to the specific sequence of each variant were designed to identify the PPARG splices expressed in the buffalo mammary gland (Appendix A). Results showed that, in early lactation, we identified the X11, X12, X13, X15, X16, X17, and X21 splices of PPARG. In middle lactation, we only obtained X3, X15, X17, and X21 splices. In late lactation, we identified X13, X15, X16, X17, and X21 splices (Figure 1A). The complete encoding sequence of the splices was further cloned from the cDNA. However, only two kinds of transcripts can be identified from the cDNA samples (Appendix A), which is consistent with the reference sequences of *PPARG-X17* and *PPARG-X21*, except for three mutational nucleotides in the obtained *PPARG-X21* sequence (Appendix A). However, they are all synonymous mutations (Ala: GCC → GCG; Ser: TCC → TCT; Val: GTC → GTG) and cannot alter the amino acid sequence. The expression level was further detected using qRT-PCR. Results showed that both *PPARG-X17* and *PPARG-X21* were highly expressed in early and middle lactation and significantly decreased in late lactation (Figure 1B). 

### 3.3. Characteristic Analysis of the PPARG Isoforms

*PPARG-X17* and *PPARG-X21* encode two proteins with 461 and 338 amino acids, respectively (Table 3). A further analysis showed that the amino acid sequences of the two protein isoforms were almost completely different, indicating that PPARG-X17 and PPARG-X21 were two completely different proteins (Appendix A). The common characteristics of the two isoforms were predicted using the ProtParam tool. Results showed that the molecular weights of the two proteins were 52.2 kDa and 38.71 kDa, and their instability coefficients were 59.53 and 49.14. The theoretical isoelectric points and the half-lives of the two proteins were 7.61 and 30 h, respectively (Table 3). Upon analyzing the secondary structure of the two proteins, we observed that random coils are the main structures in PPARG-X17, while α helices are the main structures in PPARG-X21 (Figure 2A). A prediction of the subcellular localization showed that both the PPARG-X17 and PPARG-X21 isoforms are potentially located in the nucleus (Table 3). No membrane-spanning domain nor highly hydrophobic domain were found in the PPARG-X17 and PPARG-X21 proteins (Figure 2B). Protein structure analysis also showed that the two PPARG isoforms have almost completely different folding patterns, indicating that they may have different functions or mechanisms of action. (Figure 2C). The conserved domains in the protein were further predicted using the MEME suite tool. The cladogram and structure of the PPARG genes were analyzed and are shown in Figure 3A,B. Five motifs were identified in the proteins, and PPARG-N was found to be the most conserved domain (Figure 3C,D). The PPARγ-X17 variant has two enzymatic active domains (N-terminal domain of tRNA intron endonuclease and catalytic C-terminal domain of tRNA-intron endonuclease), while PPARγ-X21 has a PPAR ligand-binding domain (NR_LBD_PPAR), which is the main functional location of PPARG (Figure 3C,D). 

### 3.4. Effect of PPARG-X17 Interference and Overexpression on Gene Expression and Fat Synthesis in BMECs

The siRNA and overexpression vector of *PPARG-X17* were transfected into the BMECs, respectively. The transfected cells showed typical cobblestone-like morphology under the microscope (Figure 4A) and showed green fluorescence when the overexpression vector was transfected (Figure 4B), indicating the successful transfection of the overexpression vector in the cells. The interference efficiency of each siRNA was analyzed, and the siRNA with the highest interference efficiency was selected for the following experiments (Figure 4C). The expression of several genes related to FA synthesis was detected using qRT-PCR. Results showed that, after RNAi of *PPARG-X17* was performed, the expression of *GPAT* (Glycerol-3-Phosphate Acyltransferase), *AGPAT6* (1-Acylglycerol-3-Phosphate O-Acyltransferase 6), *DGAT1* (Diacylglycerol O-Acyltransferase 1), *ACACA* (Acetyl-CoA Carboxylase α), *CD36* (also known as a Fatty Acid Translocase), and *ACSL1* (Acyl-CoA Synthetase Long-Chain Family Member 1) was significantly decreased (*p* < 0.05) (Figure 4D). After *PPARG-X17* overexpression, the expression of ACSL1, CD36, and GPAM was significantly increased (*p* < 0.05). However, the expression of ACACA was found to be significantly decreased (Figure 4E). The effects of RNAi and the overexpression of PPARG-X17 on the FA content and composition in the BMECs were further analyzed (Table 4). Results showed that all kinds of FAs detected in this study were significantly decreased (*p* < 0.05) after RNAi of PPARG-X17. The contents of TFAs, UFAs, SFAs, MCFAs, and LCFAs were also decreased (*p* < 0.05) compared to those in the control group (Table 4). After *PPARG-X17* overexpression, SFA content was significantly decreased (*p* < 0.05), while UFA content, especially MUFA content, was significantly increased (*p* < 0.05). However, although the total fatty acid content increased after *PPARG-X17* overexpression, there was no statistically significant difference compared with the control group (Table 4). 

### 3.5. Effect of PPARG-X21 Interference and Overexpression on Gene Expression and Fat Synthesis in BMECs 

The transfection of *PPARG-X21* siRNA and the overexpression vector was also performed and detected (Figure 5A,B). Results showed that the expression of *PPARG-X21* was almost completely silenced after RNA interference (Figure 5C). The expression of genes related to FA synthesis was also detected. Results showed that, after RNAi of *PPARG-X21* was performed, the expression of *GPAT*, *AGPAT6*, *DGAT1*, and *CD36* was significantly increased (*p* < 0.05) (Figure 5D), while the expression of ACSL1 was significantly decreased (*p* < 0.05). After *PPARG-X21* overexpression, the expression of *CD36* and *GPAM* was also found to be significantly increased (*p* < 0.05). However, the expression of *ACACA* was found to be significantly decreased (*p* < 0.05) (Figure 5E). Further analysis of the FA content and composition in the BMECs showed that the effect of *PPARG-X21* RNAi was similar to that of *PPARG-X17*, which significantly decreased all kinds of FAs in the BMECs (Table 5). Meanwhile, the FA contents in the cells after *PPARG-X17* RNAi were significantly lower (*p* < 0.05) than that performed *PPARG-X21* RNAi (Table 5), suggesting that the *PPARG- X17* had a greater effect on FA synthesis in the cells. After *PPARG-X21* overexpression, the content of total FAs and UFAs, especially MUFA content, was significantly increased (*p* < 0.05) (Table 5), while the SFA content was significantly decreased (*p* < 0.05). In addition, we found the UFAs and FAs in BMECs that overexpressed *PPARG-X21* are significantly higher (*p* < 0.05) than in those that overexpressed *PPARG-X17* (Table 5).

## 4. Discussion

PPARG is one of the most important transcriptional regulators of adipogenesis and adipocyte differentiation, which is expressed and plays essential roles in a variety of tissues [18,19]. At present, diverse PPARG transcript variants have been found in various tissues and animals. It was found that, in humans, *PPARG1* is widely expressed in adipose tissue, liver, kidney, lung, and rectum, while *PPARG2* is mainly expressed in adipose tissue and has a higher adipogenic ability [20]. There are 21 *PPARG* transcript variants with 4 non-frameshifting indels in the buffalo genomes from the NCBI database. However, little information is available about the PPARG transcript variants that are expressed in the buffalo mammary tissues. A previous study found that there are two types of *PPARG* transcripts (*PPARG1* and *PPARG2*) with 8 alternative splicing variants, among which only *PPARG1 a*, *PPARG2 a*, and *PPARG2 d* can encode complete functional proteins in buffalo mammary tissues [14]. In this study, we found that only *PPARG-X17* and *PPARG-X21* can be identified and cloned, which is inconsistent with the report. Further analysis showed that both of the *PPARG* transcripts obtained in the above report were covered by *PPARG-X15*, which was not obtained in this study. This difference may be due to the different RNA extraction methods or buffalo species used in the study. In the reference, the authors collected mammary tissues via puncture sampling from Binglangjiang buffalo [14]. In this study, we collected the mammary epithelial cells from the milk of Murrah buffalo for RNA extraction and cDNA synthesis. However, the regulatory mechanism that led to this difference needs further study. 

FA synthesis in milk is a multistep process regulated by diverse gene networks, and PPARG is considered a key signal pathway. Generally, there are four major domains involved in PPARG: the ligand-dependent transcriptional activation domain (A/B domain), DNA-binding domain (C domain), hinge domain (D domain), and ligand-binding domain (E/F domain) [19,21]. During FAs metabolism, PPARG can form heterodimers by binding with retinoid X receptor (RXR), and then it can regulate transcription and expression by binding to the PPAR response element (PPRE) of downstream genes [19]. Therefore, we further analyzed the protein characteristics and structures of PPARG-X17 and PPARG-X21. We found that only PPARγ-X21 had the RXR-binding domain (ligand-binding domain), while PPARγ-X17 had the tRNA endonuclease domain and tRNA-lyase domain. In addition, the two PPARG isoforms seem like two absolutely different proteins as they have different amino acid sequences, molecular weights, conserved domain, and secondary and tertiary structures, suggesting that their functional mechanisms may be different. However, although it does not contain an RXR-binding domain, the expression of PPARγ-X17 still significantly affected the FA synthesis in the BMECs, which is similar to PPARγ-X21. For example, after RNAi of *PPARG-X17* and *PPARG-X21*, all kinds of detected FAs, along with the total FA, UFA, SFA, MCFA, and LCFA contents in the BMECs were found to be significantly decreased. Meanwhile, both the overexpression of *PPARγ-X17* and *PPARG-X21* can significantly decrease SFA content and increase UFA content, especially MUFA content, in the BMECs. Overexpression of *PPARG-X21* significantly increased the total FA content. While *PPARγ-X17* overexpression showed an increasing trend of total FAs, although it was not significantly different from the control group (*p* = 0.055). In addition, overexpression of *PPARG-X21* appears to contribute to promoting higher function in FA synthesis than that of *PPARG-X17*. It is worth noting that both the PPARG-X17 and PPARG-X21 were located in the nucleus, with no transmembrane regions, signal peptides, nor high hydrophobic domain, which reflected the characteristics of nuclear receptors. 

The development of mammary glands and lactation is a highly coordinated process that involves an extremely complex signaling regulatory network, and PPARG is central for milk fat synthesis regulation. Many studies have been conducted on the mechanism of the PPARG gene regulating fatty acid metabolism [22]. However, few studies focus on the expression patterns of different *PPARG* splices in the buffalo mammary gland and the mechanism of regulating milk fat synthesis. Therefore, the underlying regulatory mechanism of *PPARγ-X17* and *PPARG-X21* on FA synthesis was analyzed in this study. We found that the expression of *ACACA*, *ACSL1*, *CD36*, *GPAT*, *AGPAT6*, and *DGAT* was significantly decreased after the knockdown of *PPARγ-X17* (Figure 6), suggesting that the knockdown of *PPARγ-X17* can inhibit multiple steps of fatty acid synthesis. Previous research has found that the expression of genes related to fatty acid uptake (*CD36*), de novo synthesis of fatty acids (*ACACA*, *FASN*, *SREBF1*), and triglyceride synthesis (*LPIN1*, *SCD*) was concertedly up-regulated when the bovine mammary epithelial cells (MacT cells) were treated with the PPARG-specific agonist (Rosiglitazone) [23], indicating that *PPARG* can regulate multiple steps related to fatty acid synthesis and affect the synthesis of fatty acids and triglycerides in mammary gland cells, which is consistent with our result. Among the genes, *ACACA* catalyzes the first step of the de novo synthesis of fatty acids, carboxylating acetyl-CoA to malonyl-CoA, and is a rate-limiting enzyme that regulates the de novo synthesis of fatty acids in animal tissues [24,25]. *ACSL1* and *CD36* are key proteins in the uptake and application of long-chain fatty acids from the blood by breast tissue [26,27]. The *ACSL1* has been found to play key roles in the regulation of triglyceride synthesis by activating fatty acid, i.e., altering the free LCFAs to fatty acyl-CoA esters [28]. *CD36*, also known as the scavenger receptor B2, is a key fatty acid sensor and regulator of lipid metabolism, which has a high affinity for long-chain fatty acids and can bind and transport them into cells [27,29,30]. Reports have found that the expression of *CD36* was significantly increased in the lactation of cows, and the overexpression of *CD36* increased the uptake of LCFAs in the mammary cells [31]. Triacylglycerol (TAG), which comprises 98% of the fat in buffalo milk, is synthesized from fatty acids and glycerin by enzymes such as GPAT, AGPAT, LPIN, and DGAT [32]. Among them, GPAM catalyzes glycerol-3-phosphate to produce lysophosphatidic acid, AGPAT catalyzes the conversion of LPA to phosphatidic acid, then further dephosphorylates it to form DAG, and then catalyzes it with DGAT to produce triglyceride [16]. In this study, we found that after *PPARγ-X17* overexpression, the expression of *ACSL1*, *CD36*, and *GPAM* was significantly increased, while the expression of *ACACA* was found to be significantly decreased (Figure 6), suggesting that *PPARγ-X17* overexpression can regulate fatty acid transport and triglyceride synthesis but does not affect the de novo synthesis of fatty acids.

It is worth noting that, although the regulatory effect of RNAi of *PPARG-X17* and *PPARG-X21* on FA synthesis in BMECs are similar, the gene regulatory mechanism seems different. In this study, the expression of *CD36*, *GPAT*, *AGPAT6*, and *DGAT* was significantly decreased after the knockdown of *PPARγ-X17* but significantly increased after the knockdown of *PPARγ-X21*. The reason why the opposite regulation of gene expression led to the same reduction in fatty acid synthesis is still unknown. In addition, the expression of *CD36* and *DGAT* was significantly increased both after the knockdown and overexpression of *PPARγ-X21*. The expression of *ACSL1* was significantly decreased after *PPARγ-X21* RNAi, while that of *GPAT* and *AGPAT6* was significantly increased, and the expression of *ACACA* was significantly decreased after *PPARγ-X21* overexpression. These different effects between *PPARG-X17* and *PPARG-X21* may be due to the structure differences of the proteins. However, research studies are required to give a reasonable explanation about the confusing effect of *PPARγ-X21* on the expression of FA synthesis-related genes.

## 5. Conclusions

In this study, we found only that the *PPARγ-X17* and *PPARG-X21* isoforms were expressed in the buffalo mammary gland during lactation. The PPARγ-X17 and PPARG-X21 proteins showed different characteristics and structure, with only PPARγ-X21 having an RXR-binding domain. A further analysis showed that PPARγ-X17 can modulate the FA content and composition by regulating the gene expression related to multiple steps of FA synthesis. The mechanism of *PPARγ-X21* regulating FA synthesis and metabolism seems to be very different from that of *PPARγ-X17* and needs to be further studied to reveal its specific mechanism. This study provides a further understanding of the function and mechanism of buffalo PPARG in milk fat synthesis.

## Figures and Tables

**Figure 1 genes-15-00779-f001:**
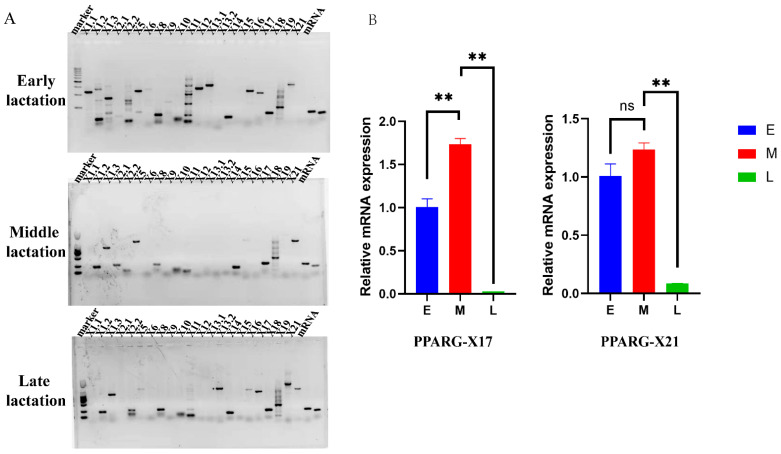
Identification of PPARG splices from buffalo somatic cells in milk. (**A**) Identification of PPARG splices from the DNA of buffalo milk somatic cells. (**B**) Expression of PPARG splices in different lactation periods; “**” represents *p* < 0.01.

**Figure 2 genes-15-00779-f002:**
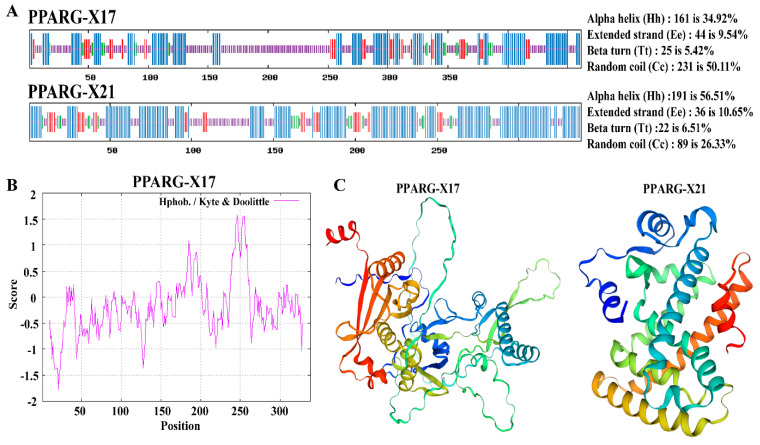
Characteristic analysis of the PPARG isoforms. (**A**) Secondary structure analysis of PPARG-X17 and PPARG-X21. (**B**) Hydrophilia analysis of PPARG-X17 and PPARG-X21. (**C**) The predicted tertiary structures of PPARG-X17 and PPARG-X21.

**Figure 3 genes-15-00779-f003:**
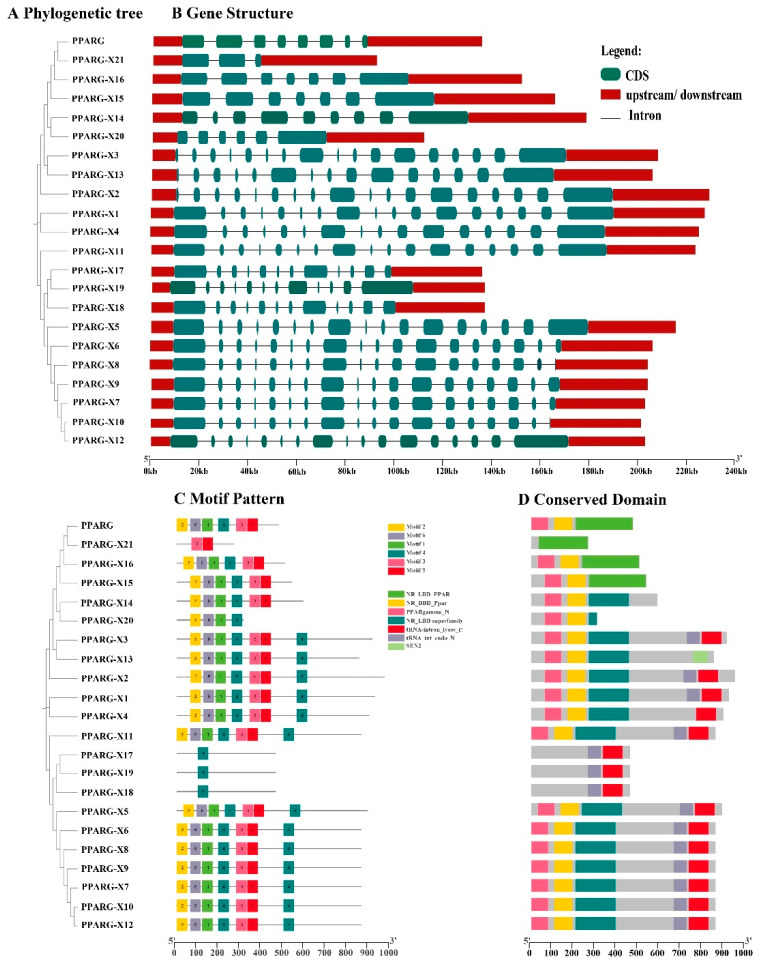
Protein domain analysis of the PPARG isoforms. (**A**,**B**) The cladogram and structure of the PPARG genes. (**C**) Motif pattern of the PPARG genes. (**D**) Conserved domain analysis of the PPARG genes.

**Figure 4 genes-15-00779-f004:**
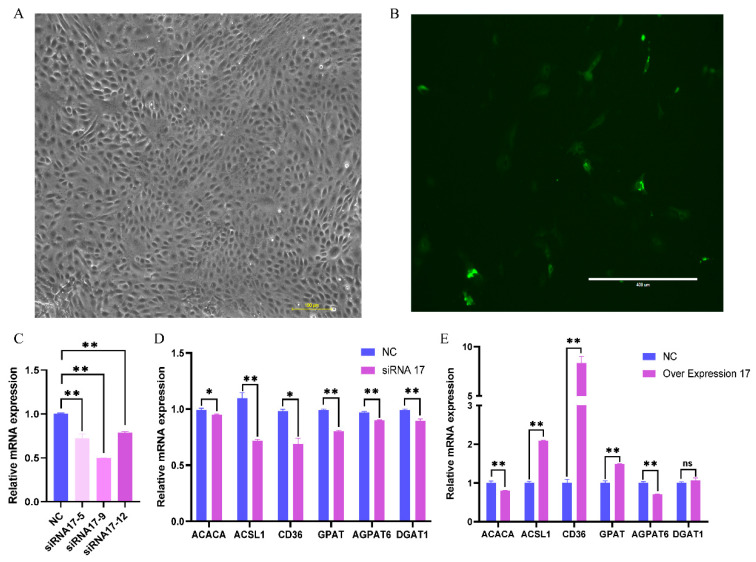
Effect of PPARG-X17 interference and overexpression in buffalo mammary epithelial cells (BMECs). (**A**) The BMECs transfected with PPARG-X17 siRNA. (**B**) The BMECs transfected with PPARG-X17 overexpression vector. (**C**) The relative expression of PPARG-X17 after siRNA interference. (**D**) Effect of PPARG-X17 RNAi on the expression of fat synthesis-related genes in BMECs. (**E**) Effect of PPARG-X17 overexpression on the expression of fat synthesis-related genes in BMECs. “*” and “**” represents *p* < 0.05 or *p* < 0.01. “ns” represents no significant difference.

**Figure 5 genes-15-00779-f005:**
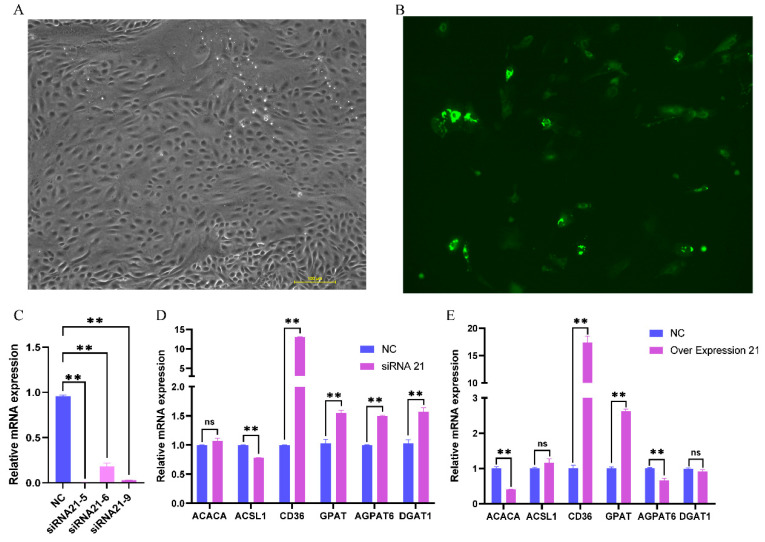
Effect of PPARG-X21 interference and overexpression in buffalo mammary epithelial cells (BMECs). (**A**) The BMECs transfected with PPARG-X21 siRNA. (**B**) The BMECs transfected with PPARG-X21 overexpression vector. (**C**) The relative expression of PPARG-X21 after siRNA interference. (**D**) Effect of PPARG-X21 RNAi on the expression of fat synthesis-related genes in BMECs. (**E**) Effect of PPARG-X21 overexpression on the expression of fat synthesis-related genes in BMECs. “**” represents *p* < 0.01. “ns” represents no significant difference.

**Figure 6 genes-15-00779-f006:**
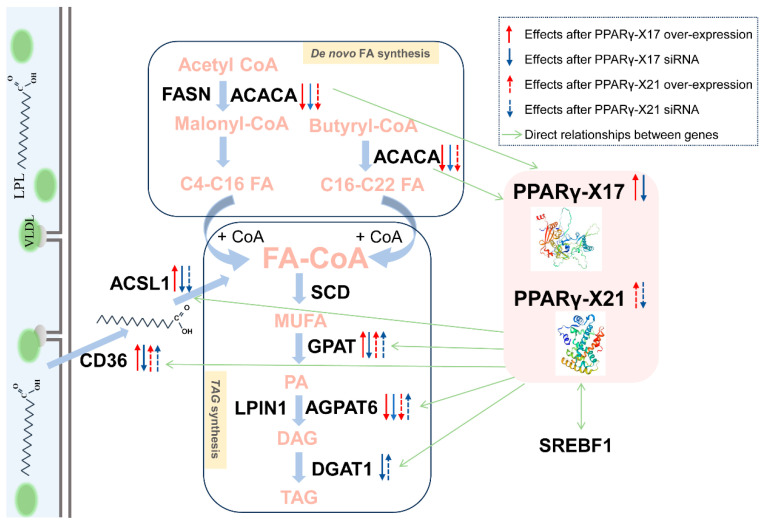
Schematic diagram shows the effect of PPARG-X17 and PPARG-X21 on fatty acids synthesis in buffalo mammary epithelial cells.

**Table 1 genes-15-00779-t001:** Routine composition of buffalo milk from different lactation periods.

Different Lactation Periods	Milk Fat (%)	Milk Protein (%)	Lactose (%)	Total Milk Solids (%)	Non-Lactose Solids (%)
Early Lactation	6.72 ± 0.46 ^a^	4.41 ± 0.02 ^a^	5.29 ± 0.11 ^a^	17.82 ± 0.55 ^a^	10.24 ± 0.21 ^a^
Mid-Lactation	10.03 ± 0.51 ^b^	4.57 ± 0.08 ^ab^	5.08 ± 0.11 ^a^	20.78 ± 0.58 ^ab^	9.74 ± 0.09 ^a^
Late Lactation	10.45 ± 0.54 ^b^	4.97 ± 0.09 ^b^	5.07 ± 0.08 ^a^	21.52 ± 1.34 ^b^	10.15 ± 0.12 ^a^

Note: Data are shown as mean ± SEM. Different superscript letters “a,b” indicate significant difference (*p* < 0.05).

**Table 2 genes-15-00779-t002:** Fatty acid composition and content in buffalo milk from different lactation periods.

Fatty Acid Content	Early Lactation	Mid-Lactation	Late Lactation
C4:0	300.0 ± 40.0	330.0 ± 80.0	330.0 ± 40.0
C6:0	100.0 ± 20.0	110.0 ± 20.0	100.0 ± 10.0
C8:0	50.0 ± 1.0	100.0 ± 1.0	40.0 ± 1.0
C10:0	80.0 ± 2.0	90.0 ± 2.0	80.0 ± 1.0
C11:0	/	20.0 ± 0.0	/
C12:0	120.0 ± 3.0	130.0 ± 3.0	110.0 ± 1.0
C13:0	5.6 ± 0.8 ^a^	7.0 ± 0.13 ^b^	7.2 ± 0.11 ^b^
C14:0	520.0 ± 10.0	600.0 ± 9.0	560.0 ± 6.0
C14:1	20.0 ± 1.0 ^a^	30.0 ± 0.01 ^b^	40.0 ± 2.0 ^b^
C15:0	60.0 ± 1.0 ^a^	80.0 ± 2.0 ^b^	80.0 ± 1.0 ^ab^
C16:0	2220.0 ± 32.0	2180.0 ± 36.0	2360.0 ± 17.0
C16:1	70.0 ± 1.0 ^a^	10.0 ± 0.0 b	20.0 ± 0.0 ^c^
C17:0	50.0 ± 1.0	60.0 ± 1.0	60.0 ± 1.0
C17:1	20.0 ± 1.0	60.0 ± 1.0	20.0 ± 0.0
C18:0	1010.0 ± 10.0	1040 ± 26.0	1000 ± 22.0
C18:1	1640.0 ± 23.0 ^a^	1840.0 ± 35.0 ^ab^	2130 ± 42.0 ^b^
C18:1 n9 c	1640.0 ± 23.0 ^a^	1820.0 ± 40.0 ^ab^	2130.0 ± 42.0 ^b^
C18:2	230.0 ± 4.0	250.0 ± 5.0	290.0 ± 6.0
C18:3	17.5 ± 2.7 ^a^	51.8 ± 1.2 ^b^	18.62 ± 3.99 ^a^
C18:2n6 t	60.0 ± 3.0	60.0 ± 1.0	80.0 ± 2.0
C18:2n6 c	170.0 ± 3.0	170.0 ± 3.0	200.0 ± 3.0
C18:3n3 (ALA)	10.0 ± 0.0	10.0 ± 0.0	10.0 ± 0.0
C18:3n6	/	10.0 ± 0.0	10.0 ± 0.0
C20:0	20.0 ± 0.0	20.0 ± 0.0	20.0 ± 0.0
C20:1	10.0 ± 0.0	10.0 ± 0.0	10.0 ± 0.0
C21:0	150.0 ± 2.0 ^a^	190.0 ± 4.0 ^b^	22.0 ± 3.0 ^b^
C22:0	10.0 ± 0.0	10.0 ± 0.0	10.0 ± 0.0
C20:3n6	10.0 ± 0.0	10.0 ± 0.0	10.0 ± 0.0
C20:3n3	3.0 ± 0.0	3.0 ± 0.1	3.0 ± 0.1
C20:4n6	9.3 ± 015	10.02 ± 0.34	12.1 ± 0.25
C21:0	150.0 ± 2.0 ^a^	190.0 ± 4.0 ^b^	22.0 ± 3.0 ^b^
C22:6n3 (DHA)	2.1 ± 0.5	1.7 ± 0.4	/
C23:0	10.0 ± 0.0	/	/
C24:0	3.5 ± 0.4 ^a^	5.0 ± 0.1 ^ab^	5.3 ± 0.1 ^b^
C24:1	/	1.8 ± 0.1	1.9 ± 0.3
C20:5n3 (EPA)	5.5 ± 0.9	3.8 ± 0.14	5.6 ± 0.38
CLA	224 ± 3.0 ^a^	235.0 ± 4.0 ^a^	281.0 ± 4.0 ^b^
Total SFA	4859.1 ± 45.0 ^a^	5162 ± 38.0 ^b^	4806.5 ± 17.0 ^ab^
Total UFA	2501.4 ± 25.6 ^a^	2747.12 ± 47.0	3142.22 ± 44.0 ^b^
Total PUFA	741.4 ± 1.6 ^a^	815.32 ± 7.0 ^b^	920.32 ± 6.0 ^b^
Total MUFA	1760 ± 24.0 ^a^	1931.8 ± 40.0 ^b^	2221.9 ± 38.0 ^b^
Total SCFA	400 ± 5.0 ^a^	440 ± 8.0 ^a^	430 ± 5.0 ^a^
Total MCFA	250 ± 13.0 ^a^	340 ± 11.0 ^b^	230 ± 9.0 ^a^
Total LCFA	6710.5 ± 93.0 ^a^	7129.12 ± 70.0 ^b^	7288.72 ± 84.0 ^b^
Total FA	7360.5 ± 70.6 ^a^	7909.12 ± 85.0 ^b^	7948.72 ± 61.0 ^b^

Note: Data are shown as mean ± SEM. “/” indicates undetectable; different superscript letters “a,b,c” indicate significant difference (*p* < 0.05). CLAs, conjugated linoleic acids. SFAs, saturated fatty acids. UFAs, unsaturated fatty acids. PUFAs, polyunsaturated fatty acids. MUFAs, monounsaturated fatty acids. SCFAs, short-chain fatty acids (C4–C6). MCFAs, midchain fatty acids (C8–C12). LCFAs, long-chain fatty acids (>C12). All milk FA compositions were expressed as mg/100 g of fat.

**Table 3 genes-15-00779-t003:** Prediction of physicochemical properties of PPARG-X17 and PPARG-X21 proteins.

	PPARG-X17	PPARG-X21
Formula	C_2319_ H_3659_ N_653_ O_689_ S_17_	C_1736_ H_2795_ N_461_ O_500_ S_18_
Number of amino acids	461	338
Molecular weights	52.2 kD	38.7 kD
Instability coefficients	59.53	49.14
Isoelectric points (PI)	7.61	7.61
Half-lives	30 h	30 h
Subcellular localization	30% located in nucleus	27% located in nucleus

**Table 4 genes-15-00779-t004:** Effect of *PPARG-X17* interference and overexpression on FA composition in BMECs.

Fatty Acid Content	Interference Control	Interference Group	Overexpression Control	Overexpression Group
C10:0	31.046 ± 0.447 ^a^	16.602 ± 1.07 ^b^	/	/
C14:0	43.861 ± 1.186 ^a^	23.167 ± 1.127 ^b^	/	/
C15:0	22.766 ± 1.029 ^a^	11.841 ± 0.521 ^b^	4.2 ± 0.781 ^a^	3.967 ± 0.252 ^b^
C16:0	122.647 ± 0.814 ^a^	70.815 ± 3.800 ^b^	103.2 ± 1.489 ^a^	46.78 ± 1.426 ^b^
C16:1	24.356 ± 1.310 ^a^	12.631 ± 0.135 ^b^	5.767 ± 0.874 ^a^	7.2 ± 0.781 ^b^
C17:0	23.926 ± 0.857 ^a^	12.599 ± 0.615 ^b^	/	/
C18:0	92.450 ± 0.605 ^a^	50.498 ± 3.34 ^b^	99.033 ± 1.935 ^a^	52.93 ± 1.903 ^b^
C18:1n9 c	88.237 ± 1.834 ^a^	49.455 ± 1.970 ^b^	32.5 ± 4.779 ^a^	42.57 ± 4.585 ^b^
C18:2n6 c	30.856 ± 2.845 ^a^	19.698 ± 0.964 ^b^	22.2 ± 4.618 ^a^	45.86 ± 4.282 ^b^
C18:3n3	25.756 ± 0.609 ^a^	13.160 ± 1.263 ^b^	8.3 ± 1.570 ^a^	7.535 ± 1.383 ^b^
C18:3n6	/	/	8.3 ± 0.170 ^a^	7.7 ± 0.7 ^a^
C20:0	48.986 ± 3.108 ^a^	24.937 ± 0.968 ^b^	17.6 ± 3.470 ^a^	12.94 ± 3.606 ^b^
C20:1	24.816 ± 1.329 ^a^	12.720 ± 0.416 ^b^	11.267 ± 1.870 ^a^	9.338 ± 1.267 ^b^
C20:2	/	/	7.4 ± 1.470 ^a^	9.964 ± 1.404 ^b^
C20:3n3	26.006 ± 2.031 ^a^	13.335 ± 0.828 ^b^	9.467 ± 1.770 ^a^	12.37 ± 1.467 ^b^
C20:3n6	28.886 ± 1.917 ^a^	15.986 ± 0.857 ^b^	15.3 ± 3.070 ^a^	10.40 ± 3.303 ^b^
C20:4n6	28.790 ± 1.993 ^a^	17.410 ± 1.212 ^b^	14.167 ± 5.270 ^a^	15.32 ± 5.167 ^b^
C20:5n3	26.076 ± 1.805 ^a^	15.110 ± 1.061 ^b^	6.5 ± 2.070 ^a^	7.360 ± 2.505 ^b^
C22:0	52.436 ± 2.440 ^a^	27.583 ± 0.936 ^b^	19.833 ± 3.670 ^a^	15.66 ± 3.833 ^b^
C22:2	23.976 ± 0.979 ^a^	12.646 ± 0.889 ^b^	8.833 ± 1.970 ^a^	8.669 ± 1.833 ^b^
C22:1n9	82.068 ± 0.577 ^a^	38.137 ± 1.569 ^b^	38.567 ± 6.870 ^a^	112.8 ± 6.567 ^b^
C22:6n3	22.526 ± 1.112 ^a^	13.060 ± 1.176 ^b^	7.7 ± 1.570 ^a^	9.865 ± 1.727 ^b^
C23:0	23.886 ± 0.981 ^a^	12.659 ± 0.361 ^b^	10.4 ± 1.670 ^a^	7.636 ± 1.434 ^b^
C24:0	/	/	21.467 ± 4.070 ^a^	13.70 ± 4.467 ^b^
C24:1	26.936 ± 1.419 ^a^	15.060 ± 2.756 ^b^	12.067 ± 2.070 ^a^	8.060 ± 2.067 ^b^
CLA	30.856 ± 2.845 ^a^	19.698 ± 0.964 ^b^	22.2 ± 4.618 ^a^	45.86 ± 4.282 ^b^
Total SFA	459.387 ± 4.704 ^a^	250.361 ± 7.955 ^b^	284.7 ± 49.08 ^a^	160.2 ± 7.275 ^b^
Total UFA	35.412 ± 1.588 ^a^	19.268 ± 0.950 ^b^	208.333 ± 3.350 ^a^	314.667 ± 8.715 ^b^
Total MUFA	155.133 ± 5.843 ^a^	80.618 ± 7.011 ^b^	100.167 ± 14.054 ^a^	179.6 ± 2.961 ^b^
Total PUFA	295.670 ± 3.765 ^a^	169.864 ± 5.471 ^b^	108.167 ± 2.000 ^a^	135.067 ± 8.844 ^b^
MCFA	32.113 ± 1.875 ^a^	16.602 ± 1.07 ^b^	/	/
LCFA	887.629 ± 22.134 ^a^	484.241 ± 26.722 ^b^	379.867 ± 6.589 ^a^	416.967 ± 13.317 ^a^
Total FA	921.289 ± 24.091 ^a^	499.109 ± 27.792 ^b^	379.867 ± 6.589 ^a^	416.967 ± 13.317 ^a^

Note: Data are shown as mean ± SEM. “/” indicates undetectable. Different superscript letters “a,b” indicate significant difference (*p* < 0.05). Interference control indicates cells transfected with random siRNA sequences. Overexpression control indicates cells transfected with blank vector. CLA indicates conjugated linoleic acid. SFA indicates saturated fatty acid. UFA indicates unsaturated fatty acid. PUFA indicates polyunsaturated fatty acid. MUFA indicates monounsaturated fatty acid. SCFA indicates short-chain fatty acid (C4–C6). MCFA indicates midchain fatty acid (C8–C12). LCFA indicates long-chain fatty acid (>C12). All milk FA compositions were expressed as mg/100 g of fat.

**Table 5 genes-15-00779-t005:** Effect of *PPARG-X21* interference and overexpression on FA composition in BMECs.

Fatty Acid Content	Interference Control	Interference Group	Overexpression Control	Overexpression Group
C10:0	31.046 ± 0.447 ^a^	21.216 ± 1.603 ^b^	/	/
C14:0	43.861 ± 1.186 ^a^	29.979 ± 2.247 ^b^	/	/
C15:0	22.766 ± 1.029 ^a^	15.398 ± 0.968 ^b^	4.2 ± 0.78 ^a^	4.233 ± 0.416 ^a^
C16:0	122.647 ± 0.814 ^a^	84.563 ± 1.214 ^b^	103.2 ± 1.489 ^a^	50.03 ± 1.747 ^b^
C16:1	24.356 ± 1.310 ^a^	16.479 ± 0.90 ^b^	5.767 ± 0.874 ^a^	8.166 ± 0.472 ^b^
C17:0	23.926 ± 0.857 ^a^	15.700 ± 0.886 ^b^	/	/
C18:0	92.450 ± 0.605 ^a^	66.331 ± 3.499 ^b^	99.033 ± 1.935 ^a^	57.9 ± 2.883 ^b^
C18:1n9 c	88.237 ± 1.834 ^a^	60.896 ± 0.281 ^b^	32.5 ± 4.779 ^a^	49.53 ± 2.650 ^b^
C18:2n6 c	30.856 ± 2.845 ^a^	25.310 ± 1.633 ^b^	22.2 ± 4.618 ^a^	50.1 ± 3.934 ^b^
C18:3n3	25.756 ± 0.609 ^a^	15.904 ± 1.186 ^b^	8.3 ± 1.570 ^a^	8.166 ± 0.709 ^a^
C18:3n6	/	/	8.3 ± 0.170 ^a^	8.366 ± 0.650 ^a^
C20:0	48.986 ± 3.108 ^a^	31.863 ± 2.029 ^b^	17.6 ± 3.470 ^a^	13.63 ± 1.514 ^b^
C20:1	24.816 ± 1.329 ^a^	16.880 ± 1.207 ^b^	11.267 ± 1.870 ^a^	9.966 ± 0.929 ^a^
C20:2	/	/	7.4 ± 1.470 ^a^	10.83 ± 1.150 ^b^
C20:3n3	26.006 ± 2.031 ^a^	16.432 ± 0.217 ^b^	9.467 ± 1.770 ^a^	14.16 ± 0.665 ^b^
C20:3n6	28.886 ± 1.917 ^a^	19.339 ± 0.57 ^b^	15.3 ± 3.070 ^a^	11.76 ± 0.450 ^a^
C20:4n6	28.790 ± 1.993 ^a^	19.182 ± 0.712 ^b^	14.167 ± 5.270 ^a^	19.9 ± 1.916 ^a^
C20:5n3	26.076 ± 1.805 ^a^	17.788 ± 1.852 ^b^	6.5 ± 2.070 ^a^	7.833 ± 0.737 ^a^
C22:0	52.436 ± 2.440 ^a^	34.984 ± 1.702 ^b^	19.833 ± 3.670 ^a^	17.03 ± 0.850 ^a^
C22:2	23.976 ± 0.979 ^a^	17.038 ± 1.482 ^b^	8.833 ± 1.970 ^a^	9.166 ± 0.873 ^a^
C22:1n9	82.068 ± 0.577 ^a^	56.205 ± 1.307 ^b^	38.567 ± 6.870 ^a^	134.5 ± 3.538 ^b^
C22:6n3	22.526 ± 1.112 ^a^	16.194 ± 0.132 ^b^	7.7 ± 1.570 ^a^	9.3 ± 0.557 ^a^
C23:0	23.886 ± 0.981 ^a^	15.554 ± 0.605 ^b^	10.4 ± 1.670 ^a^	7.766 ± 0.850 ^b^
C24:0	/	/	21.467 ± 4.070 ^a^	14.33 ± 1.553 ^b^
C24:1	26.936 ± 1.419 ^a^	18.079 ± 0.491 ^b^	12.067 ± 2.070 ^a^	8.4 ± 0.656 ^b^
CLA	30.856 ± 2.845 ^a^	25.310 ± 1.633 ^b^	22.2 ± 4.618 ^a^	50.1 ± 3.934 ^b^
Total SFA	459.387 ± 4.704 ^a^	318.817.173 ^b^	284.7 ± 49.08 ^a^	172.1 ± 10.04 ^b^
Total UFA	35.412 ± 1.588 ^a^	24.704 ± 1.778 ^b^	208.333 ± 3.350 ^a^	360.233 ± 1.156 ^b^
Total MUFA	155.133 ± 5.843 ^a^	104.602 ± 7.469 ^b^	100.167 ± 14.054 ^a^	210.633 ± 4.010 ^b^
Total PUFA	295.670 ± 3.765 ^a^	207.634 ± 5.486 ^b^	108.167 ± 2.000 ^a^	149.6 ± 9.700 ^b^
MCFA	32.113 ± 1.875 ^a^	21.216 ± 1.603 ^b^	/	/
LCFA	887.629 ± 22.134 ^a^	609.838 ± 24.570 ^b^	379.867 ± 6.589 ^a^	469.9 ± 19.487 ^b^
Total FA	921.289 ± 24.091 ^a^	631.314 ± 26.173 ^b^	379.867 ± 6.589 ^a^	469.9 ± 19.487 ^b^

Note: Data are shown as mean ± SEM. “/” indicates undetectable. Different superscript letters “a,b” indicate significant difference (*p* < 0.05). Interference control indicates cells transfected with random siRNA sequences. Overexpression control indicates cells transfected with blank vector. CLA indicates conjugated linoleic acid. SFA indicates saturated fatty acid. UFA indicates unsaturated fatty acid. PUFA indicates polyunsaturated fatty acid. MUFA indicates monounsaturated fatty acid. SCFA indicates short-chain fatty acid (C4–C6). MCFA indicates midchain fatty acid (C8–C12). LCFA indicates long-chain fatty acid (>C12). All milk FA compositions were expressed as mg/100 g of fat.

## Data Availability

The dates generated and analyzed during this study are included in this paper. Additional datasets used and/or analyzed during this study are available from the corresponding author upon reasonable request.

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
