# Peer review of "A Characterization and Functional Analysis of Peroxisome Proliferator-Activated Receptor Gamma Splicing Variants in the Buffalo Mammary Gland"

_genes, 2024, doi:10.3390/genes15060779_

Round 1

Reviewer 1 Report

Comments and Suggestions for Authors

A manuscript submitted for review entitled: “Characterization and Functional Analysis of Peroxisome Proliferator-Activated Receptor Gamma Splicing Variants in Buffalo Mammary Gland” aims to demonstrate the role of PPARG in buffalo milk fat synthesis. Buffalo milk is a valuable food source for many regions of the world, which underlines the relevance of the study.

I have the following notes and recommendations for the authors:

In table one, the amount of milk from the examined animals should also be indicated.

In Tables 2 and 5 as well as in the text, the abbreviations used for SCFA, MCFA, LCFA, CLA should be described.

In my opinion, it is appropriate for the authors to hypothesize after the discovery that only PPARG-X17 and PPARG-X21 are expressed, what health properties it may lead to in humans. Positive or negative, which will be the most valuable part of the research in my opinion.

Author Response

Dear reviewers

Thanks for your letter concerning our manuscript. These comments are all valuable and very helpful for improving our manuscript, as well as the important guiding significance to our further research. We have studied all comments carefully and revised our manuscript word by word. All of the suggestions have been revised and highlighted in the manuscript. The responses to the comments are listed in the following point-by-point.

Sincerely.

Response to Reviewer 1:

  1. In table one, the amount of milk from the examined animals should also be indicated.

Reply: Thanks for your comment. We feel sorry that we did not collect the data on the amount of milk during this study. We tried to ask the farm to provide this data, but the farm only recorded the milk yield according to the whole lactation period. Therefore, we cannot obtain the data on the amount of milk at this time. We do appreciate you for providing us with a novel perspective on experimental design and hope that this explanation is informative and sufficient for you. If you have any other questions, we are always glad to learn from you.

  1. In Tables 2 and 5 as well as in the text, the abbreviations used for SCFA, MCFA, LCFA, CLA should be described.

Reply: Thanks for your comment. We have supplemented the whole name of the used abbreviations in the manuscript.

  1. In my opinion, it is appropriate for the authors to hypothesize after the discovery that only PPARG-X17 and PPARG-X21 are expressed, what health properties it may lead to in humans. Positive or negative, which will be the most valuable part of the research in my opinion.

Reply: Thanks for your comment. According to the published studies, the splice type of PPARG gene expressed in different species or tissues may be different. In this study, we focused on the effect of PPARG gene expression on fatty acid synthesis in the buffalo mammary gland. In the results, we showed that overexpression of PPARG-X17 and PPARG-X21 significantly increased the synthesis of UFA and decreased the SFA. Meanwhile, we found that overexpression of PPARG-X21 can induce more synthesis of UFA and FA than PPARG-X17. Considering UFA is beneficial to human health, we think PPARG-X21 may provide more positive health properties. However, as no human health-related detection was performed in this study, we cannot speculate on the exact relationship between this gene and the health properties in humans. We do appreciate you for providing us with a novel perspective on our further study. We hope this answer can meet your requirements and are glad to improve it again if you can give more detailed comments.

Reviewer 2 Report

Comments and Suggestions for Authors

The manuscript present sound and novel data on the buffallo mammary gland fatty acid synthesis regulator proteins PPARG_X17 and _X21. The methods are adequate and up-to-date. The discussion could have been better organized, for example a figure which descibes the role of the two proteins in FA synthesis, indicating the up and down regulated players could be helpfull. The contradiction with the former publication on PPARG proteins (12) is explained with the different types of buffalo from where the samples were originated is unlikely. The  limitations of the methods which were applied in the two sets of experiments might be the reason? It is not clear why the authors found unexpected that the two proteins showed different regulator functions, when they clearly showed that they have different functional domains, aa sequences etc!

Minor comments:

Please add the whole name of the proteins where they show up first time in the manuscript.

line 68 in cow mammary gland

line  403-04 Many studies... add a reference e.g. a review

line 440 increased, and

line 446 synthesis, few studies

Comments on the Quality of English Language

Nice piece of work with original, trustable results. After corrections it is well worth to be published

Author Response

Dear reviewers

Thanks for your letter concerning our manuscript. These comments are all valuable and very helpful for improving our manuscript, as well as the important guiding significance to our further research. We have studied all comments carefully and revised our manuscript word by word. All of the suggestions have been revised and highlighted in the manuscript. The responses to the comments are listed in the following point-by-point.

Sincerely.

Response:

The manuscript present sound and novel data on the buffallo mammary gland fatty acid synthesis regulator proteins PPARG_X17 and _X21. The methods are adequate and up-to-date.

a: The discussion could have been better organized, for example a figure which descibes the role of the two proteins in FA synthesis, indicating the up and down regulated players could be helpful.

Reply: Thanks for your comment. We have rewritten the discussion section and supplemented a figure describing the role of the two proteins in FA synthesis in the manuscript.

b: The contradiction with the former publication on PPARG proteins (12) is explained with the different types of buffalo from where the samples were originated is unlikely. The limitations of the methods which were applied in the two sets of experiments might be the reason?

Reply: Thanks for your comment. By rereading the reference (12), we agree with the reason you provided. We do appreciate you for providing us with a novel perspective regarding the discussion. We have revised the description in the manuscript.

c: It is not clear why the authors found unexpected that the two proteins showed different regulator functions, when they clearly showed that they have different functional domains, aa sequences etc!

Reply: Thanks for your comment. We want to say that, although the PPARG-X17 and PPARG-X21 come from the same gene family, the amino acid sequences, molecular weights, conserved domain, and secondary and tertiary structure are quite different. However, they showed a similar effect on the FA synthesis in BMECs. For example, after RNAi of PPARG-X17 and PPARG-X21, all kinds of detected FAs, along with the total FA, UFA, SFA, MCFA, and LCFA in the BMECs were found significantly decreased. Meanwhile, both the overexpression of PPARγ-X17 and PPARG-X21 can significantly decrease the SFA content and increase the UFA, especially the MUFA content, in the BMECs. We have revised the description in the discussion section.

Minor comments:

  1. Please add the whole name of the proteins where they show up first time in the manuscript.

Reply: Thanks for your comment. We have revised this problem according to your comments in the manuscript.  

  1. line 68 in cow mammary gland; line 440 increased, and; line 446 synthesis, few studies

Reply: Thanks for your comment. We have revised the description of these sentences in the manuscript. It should be pointed out that, in line 68, adding “cow” here is not suitable as one of the above references is about buffalo study. The highlight comparison here should be between “the expression pattern” and “the effects of PPARG on FA synthesis and regulatory functions”. We hope this answer can meet your requirements and are glad to improve it again.

  1. line 403-04 Many studies... add a reference e.g. a review

Reply: Thanks for your comment. We have supplemented a reference in the manuscript.

Reviewer 3 Report

Comments and Suggestions for Authors

Dear authors, respectfully, it seems to me that this is a good manuscript and it has relevance in the scientific world. However, many points affect the quality of the manuscript.

Abstract: This is the first printing of his manuscript; consequently, the quality of your manuscript depends on it. I like this abstract; however, in the current situation, the abstract needs improvement by adding highlighted results with their respective p-value and a specific conclusion correlated with the objective.

Introduction: Dear authors, the introduction is well written. Many good ideas are described; however, my suggestion is to add values to improve it. Only as an example: 70% of the fatty acids in milk come from “the novo synthesis”.

Lines 49-52: Add references.

Line 52: How many? Add % value(s).

Material and methods topic: Dear authors, if possible, add the dry matter, protein and fat content of the animals’ diet.

Line 79: Add the animals breed. How many times was tissue collected and at what specific time?

Results: Try using an additional writing style to highlight your results on this topic. The topic of results is not a direct translation of the tables; you can use presentation in other ways, such as proportions, ratios, etc.

Line 211: Remove this line because it is the same description as the table title.

Lines 212-223: Add values, p-value, etc. Is higher, is lower, are similar: those expressions are very generic. Similar comment for all the results description.

Figure 1: My suggestion is to separate the figures and add them to the text separately, or present them in high-quality as complementary material. Same suggestion for all the figures thought the text.

Discussion: The discussion should focus on explaining how the results were obtained and you obtained many results; however, and few was discussed. To improve the discussion, add theories, hypotheses or statements with their respective reference about how you obtained your results, whether biologically, metabolically, physiologically, environmentally, etc. In the current situation, the discussion is a good general review; however, you need to make a specific description of how the results were obtained.

Lines 357-397: This is a very long paragraph; separate it into at least 3 paragraphs.

Lines 357-377: This information should be considered in the introduction of the manuscript.

Line 381: Whit what report?

Lines 381-383: This is a discussion about why you didn't find the same answer as another author's report.

Lines 403-406: What are these manuscripts? Add the references.

Lines 428-433: this is a results description.

Lines 441-442: Speculative.

Conclusion: lines 469-471: It's true; however, this has not been highlighted in the description of the results or in the description of the discussion.

References: references must be updated; Approximately 40% of them are from the last 10 years. 10% should be the maximum value for references older than 10 years.

Author Response

Dear reviewer,

Thanks for your letter concerning our manuscript. These comments are all valuable and very helpful for improving our manuscript, as well as the important guiding significance to our further research. We have studied all comments carefully and revised our manuscript word by word. All of the suggestions have been revised and highlighted in the manuscript. The responses to the comments are listed in the following point-by-point.

Sincerely.

Response:

  1. Abstract: This is the first printing of his manuscript; consequently, the quality of your manuscript depends on it. I like this abstract; however, in the current situation, the abstract needs improvement by adding highlighted results with their respective p-value and a specific conclusion correlated with the objective.

Reply: Thanks for your comment. We have supplemented the highlighted results with their respective p-value and a specific conclusion in the abstract of the manuscript. The revised part is pasted below.

RNA interference (RNAi) and overexpression of PPARG-X17 and PPARG-X21 in buffalo mammary epithelial cells (BMECs) were performed. Results showed that the expression of fatty acid synthe-sis-related genes (ACACA, CD36, ACSL1, GPAT, AGPAT6, DGAT1) was significantly modified (P< 0.05) by the RNAi and overexpression of PPARG-X17 and PPARG-X21. All kinds of FA detected in this study were significantly decreased (P< 0.05) after RNAi of PPARG-X17 or PPARG-X21. Overexpression of PPARG-X17 or PPARG-X21 significantly decreased (P< 0.05) the SFA content, while significantly increased (P< 0.05) the UFA, especially the MUFA in the BMECs. In conclusion, there are two PPARG splicing variants expressed in the BMECs and can regulate the FA synthesis by altering the expression of diverse fatty acid synthesis-related genes.

  1. Introduction: Dear authors, the introduction is well written. Many good ideas are described; however, my suggestion is to add values to improve it. Only as an example: 70% of the fatty acids in milk come from “the novo synthesis”.

Reply: Thanks for your suggestion. We have supplemented the values we can get in the description in the manuscript. However, some of the references did not give the exact values of the results. Therefore, we retained some of the descriptions to consist of the references. We hope this answer can meet your requirements and are glad to improve it again if you can give more detailed comments.

  1. Lines 49-52: Add references.

 Reply: Thanks for your comment. We have supplemented the references in the manuscript.

  1. Line 52: How many? Add % value(s).

 Reply: Thanks for your comment. We have supplemented the value in the manuscript. 57.8 % FA in milk is derived from de novo synthesis by mammary epithelial cells from the second month of lactation.

  1. Material and methods topic: Dear authors, if possible, add the dry matter, protein and fat content of the animals’ diet.

Reply: Thanks for your comment. We didn’t measure the nutritional content of the diet of the buffalo. By asking the farmer on the farm, we obtained the feed formula and pasted it below.

Corn 30%, Alfalfa 20%, Bran 12%, Soya-bean 30%, Fish meal 5%, Bone meal 1% and CaCO3 1%.

  1. Line 79: Add the animals breed. How many times was tissue collected and at what specific time?

 Reply: Thanks for your comment. The animal breed used in this study is the Murrah buffalo. The buffalo mammary gland tissues were collected 2 times on the farm during the mid-lactation period. We have supplemented this information in the manuscript.

  1. Results: Try using an additional writing style to highlight your results on this topic. The topic of results is not a direct translation of the tables; you can use presentation in other ways, such as proportions, ratios, etc.

 Reply: Thanks for your comment. We have rewritten the related description in the manuscript and pasted it below.

Results showed that the total FA (TFA) in middle and late lactation was significantly higher than that in early lactation (P < 0.05), which is consistent with the routine analysis shown above. Among the FAs, C16:0 was the most abundant, accounting for 30.2%, 27.6%, and 29.7% of the total FA in the three periods, respectively. The content of unsaturated fatty acid (UFA), especially monounsaturated fatty acid (MUFA) is significantly increased (P < 0.05) as the lactation progresses. The UFA accounts for 33.9%, 34.7%, and 39.5% of the total FA, and the MUFA account for 23.9%, 24.4% and 27.9% of the total FA in the three periods, respectively. Docosahexaenoic acid (DHA) content is highest in milk from early lactation (2.1±0.5) and then gradually decreases in mid-lactation (1.7±0.4), and out of detection in milk from late lactation. The long-chain fatty acids (LCFA) seem to be the main increased FA (P < 0.05) as the lactation progresses (Table 1).

  1. Line 211: Remove this line because it is the same description as the table title.

 Reply: Thanks for your comment. We have deleted this line in the manuscript.

  1. Lines 212-223: Add values, p-value, etc. Is higher, is lower, are similar: those expressions are very generic. Similar comment for all the results description.

 Reply: Thanks for your comment. We have supplemented the p-value in the manuscript.

  1. Figure 1: My suggestion is to separate the figures and add them to the text separately, or present them in high-quality as complementary material. Same suggestion for all the figures through the text.

 Reply: Thanks for your comment. We have divided Figure 1 and Figure 2 into several figures and put part of them in the supplementary materials.

  1. Discussion: The discussion should focus on explaining how the results were obtained and you obtained many results; however, and few was discussed. To improve the discussion, add theories, hypotheses or statements with their respective reference about how you obtained your results, whether biologically, metabolically, physiologically, environmentally, etc. In the current situation, the discussion is a good general review; however, you need to make a specific description of how the results were obtained.

 Reply: Thanks for your comment. According to your suggestion, we have rewritten the whole discussion section in the manuscript. We do appreciate your suggestion and hope the revised discussion can meet your requirements. We are glad to improve it again if you can give more detailed comments.

  1. Lines 357-397: This is a very long paragraph; separate it into at least 3 paragraphs. Lines 357-377: This information should be considered in the introduction of the manuscript. Line 381: Whit what report? Lines 381-383: This is a discussion about why you didn't find the same answer as another author's report. Lines 403-406: What are these manuscripts? Add the references. Lines 428-433: this is a results description. Lines 441-442: Speculative.

Reply: Thanks for your comment. The whole discussion section in the manuscript has been rewritten according to all of your comments. Considering that these comments are all in the discussion section, we reply to them here together. We hope this answer can meet your requirements and are glad to improve it again if you can give more detailed comments.

  1. Conclusion: lines 469-471: It's true; however, this has not been highlighted in the description of the results or in the description of the discussion.

 Reply: Thanks for your comment. We have revised this sentence in the manuscript.

  1. References: references must be updated; Approximately 40% of them are from the last 10 years. 10% should be the maximum value for references older than 10 years.

Reply: Thanks for your comment. We have replaced most of the references older than 10 years with recent references. Now, only 4 of 32 are from the last 10 years.

Round 2

Reviewer 3 Report

Comments and Suggestions for Authors

Dear authors, I am satisfied with your answers. I would like to get a bigger improvement with my specific suggestions; however, I see that the manuscript has many generic paragraphs. Add numbers if you can. There may not be exact numbers you can find, but you can find minimums and maximums in the literature.